# Factors Facilitating the Implementation of the Sustainable Development Goals in Regional and Local Planning—Experiences from Norway

**Kjersti Granås Bardal \*, Mathias Brynildsen Reinar, Aase Kristine Lundberg** 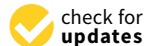 **and Maiken Bjørkan**

Nordland Research Institute, N-8049 Bodø, Norway; mbr@nforsk.no (M.B.R.); akl@nforsk.no (A.K.L.); mbj@nforsk.no (M.B.)

\* Correspondence: kgb@nforsk.no

**Abstract:** Successful implementation of the Sustainable Development Goals (SDGs) depends on regional and local authorities' ability to implement the goals in their respective contexts. Through a survey and interviews with informants in Norwegian municipalities and county councils, this paper explores and offers new empirical insight into (1) which factors can be identified as facilitating the implementation of the SDGs in Norwegian local and regional planning; (2) how the facilitating factors are conditioned by the different local and regional institutional contexts; and (3) how these factors from the Norwegian context correspond or differ from those in the international literature. We find that the existing Planning and Building Act is considered a suitable framework for the implementation of the SDGs in the Norwegian context, and that the SDGs are high on the national and regional governmental agendas. However, work remains in integrating the SDGs into underlying governmental activities. They must be incorporated into action plans and planning tools, which will require involvement, collaboration and development work across sectors and authority levels, and the development of guidelines for how this can be done. Allocating enough resources for this work will be crucial, and smaller municipalities may need other types and degrees of support than larger ones.

**Keywords:** sustainable development goals; facilitating factors; implementation; regional and local planning; Norway

## 1. Introduction

Sustainable development has been a guiding norm and political objective ever since the Brundtland Commission published the report Our Common Future in 1987. The most frequently quoted definition of sustainable development is from this report [1] (p. 43): "Sustainable development is development that meets the needs of the present without compromising the ability of future generations to meet their own needs". The concept of sustainable development aims to maintain economic advancement and progress while protecting the long-term value of the environment [2]. In a common understanding of the concept, both economic, social and environmental aspects need to be integrated in decision making and balance each other."

In 2015, the United Nation's General Assembly further followed up the Brundtland report and adopted the 2030 Agenda and Sustainable Development Goals (SDGs). The agenda, with its 17 SDGs and 169 targets, is a universal call to action to end poverty, protect the planet and ensure that all people enjoy peace and prosperity by 2030 [3]. The 17 SDGs are integrated and recognize that development must balance social, economic and environmental sustainability. In order to reach the ambitious agenda within this decade, all parts of society, in all countries and regions, must contribute. Although the 2030 Agenda and the SDGs have relatively newly been adopted, an extensive literature that

concerns various topics related to them has already emerged—see for instance Alibašić [4], Monkelbaan [5] and Nhamo et al. [6].

As a holistic framework, the SDGs challenge actors across different levels and sectors not only to understand how they influence the prosperity of people and planet, but to act to progress towards a more sustainable and just world. Even though the SDGs have been adopted at the supranational level in the UN, implementation has to be bottom-up. It has been estimated that as much as 65 percent of the targets cannot fully be achieved without the involvement of local actors [7]. Through the concept of localizing the SDGs, the central role of local authorities, civil society organizations and other local stakeholders has been recognized [8]. Thus, successful implementation of the SDGs depends on national, regional and local authorities' ability to translate the goals and targets into their respective contexts, and their ability to implement measures that ensure a holistic approach to the SDGs [9].

Countries and regions differ when it comes to geographic, demographic and economic situations, as well as their legal, democratic and governing systems. This again impacts what national, regional and local authorities experience as challenges when putting the SDGs into action [10,11]. Advantages, conflicts and tensions occurring in the localization process will be formed by the various economical, institutional, social and cultural territories in which the localization process is embedded [12]. Scholars have argued for a greater emphasis on the territorial embeddedness and multi-scalar nature of sustainability transitions, since this can enable a richer understanding of the different ways spatial contexts shape transition processes and the multiplicity and heterogeneity of transition pathways [12]. Furthermore Kulonen et al. [13] emphasize the scientific and political motivations for spatial considerations for sustainable development and that spatial dimensions need to be accounted for in the SDGs framework.

The adoption in 2015 of Agenda 2030 and the 17 UN SDGs has, therefore, led to a growing literature describing factors that either facilitate or challenge the implementation of the SDGs in various contexts.

In 2020, Norway was ranked as the sixth country in the world on overall SDG score, measuring countries' total progress toward archiving all 17 SDGs [14]. Internationally, Norway was an advocate for adopting Agenda 2030, while the Norwegian prime minister headed the UN-appointed SDG advocate group. In the national expectations for local and regional planning, the government stated in 2019 that the SDGs should be the main political framework to address the greatest challenges of our time, such as poverty, climate change and inequality. Further, they underlined that the regional and local authorities' efforts are crucial for Norway's contribution to meeting Agenda 2030, since they are closest to people, local businesses and organizations. The government also emphasized that local and regional authorities are responsible for much of the social and physical infrastructure impacting peoples living conditions and opportunities for development. Therefore, the government expects the SDGs to be implemented and become a foundational part of regional and local planning [15] (p. 3).

In Norway, municipalities (total of 356) have the principal authority to make decisions about land use. However, both regional and national authorities have a say in these processes and seek to influence local planning through national expectations, guidelines and planning provision. At the regional level, county councils are responsible for ensuring holistic and coordinated planning across the municipalities but have no formal authority to dictate local planning. Rather, the counties seek to influence local planning through contributing knowledge, advice and guidance, but also facilitating networks and arenas for local planners, which is particularly important in small municipalities.

That both social and spatial planning should play a central role in delivering sustainable development is not new in the Norwegian context. The purpose of the Norwegian Planning and Building Act of 2008 (PBA) is to "promote sustainable development in the best interests of individuals, society and future generations." However, evaluations of how the PBA works in practice show that it is difficult to balance the three dimensions of sustainability and it has proven difficult to integrate the inter- and intra-generational

perspective to sustainability in specific planning decisions [16]. At the same time, previous studies have shown that many small and rural municipalities in Norway have challenges related to resources and capacity when it comes to planning [17].

Despite the underlining of the importance of commitment and effort by county councils and municipalities for Norway's contribution to meeting Agenda 2030, the Norwegian Auditor General [18] recently criticized the government for not coordinating the national implementation of the SDGs sufficiently. The lack of a comprehensive national plan for implementation has resulted in a fragmented approach, in stark contrast to the holistic and cross-sectoral approach needed to realize Agenda 2030. In turn, this has also affected the pace of the national implementation, which the Auditor General describes as being behind other Nordic countries. With this as a backdrop, Norway is an interesting case to examine experiences at the local and regional level with the implementation of the SDGs.

With this article, we aim to expand the existing literature on factors facilitating the implementation of the SDGs in local and regional planning, by providing new empirical evidence from Norwegian municipal and county administrations. Local and regional planning covers both rural and urban municipalities and regions of various size. We focus on the institutional context in which the implementation process takes place. In Norway the various municipal administrations and county councils differ in how far they have come in implementing the sustainable development goals in planning [9]. Thus, our hypotheses are that there are various factors that facilitates or hinders the implementation process, and these factors are affected by institutional contexts among local and regional authorities. The research questions explored are, therefore, as follows:

1. What factors can be identified as facilitating the implementation of the SDGs in Norwegian local and regional planning?
2. How are the facilitating factors conditioned by the different local and regional institutional contexts?
3. How do these factors from the Norwegian context correspond to or differ from those in the international literature?

We understand facilitating factors broadly as key factors for succeeding with the implementation of policies such as the SDGs in local and regional planning. They are success factors dealing with potential barriers related to the implementation. The factors may be seen as products of the local context, the type of policy in question, and the policy process [19].

Data have been collected through both an electronic survey among professionals working with planning and/or environmental issues in Norwegian municipalities and through semi-structured interviews with key informants from six municipalities and five county councils in Norway.

By contributing with new empirical knowledge, this study provides valuable learning for local, regional and national authorities, politicians and the academic field working with the implementation of the SDGs. Knowledge about facilitating factors can contribute to reinforce and strengthen the capacity of local and regional governments to deliver on Agenda 2030. Furthermore, understanding the facilitating factors of local SDG implementation is vital for stepping up the pace in the coming decade.

The article is structured as follows. In Section 2 we present our analytical framework for studying factors facilitating the implementation of SDGs. The data and methods are described in Section 3. In Section 4 we present the results and discuss them in relation to the three research questions. The article ends with some concluding remarks in Section 5.

## 2. Analytical Framework: Factors Facilitating the Implementation of the SDGs

Policy implementation is what develops between the establishment of an apparent intention on the part of government to do something, or to stop doing something, and the ultimate impact in the world of action [20]. Policy implementation reflects a complex change process where government decisions are transformed into programs, procedures, regulations, or practices with various aims [21]. Implementation of the SDGs in local and

regional planning will involve such complex change processes. Tools and guidance on how to localize the SDGs and implement them in planning have been developed by, for example, the Global Taskforce of Local and Regional Governments [22]. Alibašić [4] describes how local governments can design and implement sustainability policies, initiatives and programs by offering guidance, strategies, and methods in applying sustainability and resilience planning, while Nhamo et al. [6] show how to draw national level baselines for the localization of the SDGs, aiming to provide a clear roadmap toward achieving Agenda 2030. The authors are cognizant of various institutions' common but differentiated responsibilities and capabilities within their socio-political, environmental, and economic conditions.

Implementation of the SDGs can be both hindered and facilitated by various factors. DeGroff and Cargo [21] identify three factors affecting policy implementation processes that they argue are of particular importance: networked governance, socio-political context, and democratic factors. Other literature also emphasizes how contextual and cultural influences affect policy implementation in significant ways, and that the effectiveness of any policy is also shaped and molded by context and culture [23].

Factors influencing policy implementation may be categorized in various ways. We have been inspired by the framework of Åkerman et al. [24] on barriers and success factors, and have categorized the factors facilitating the implementation of the SDGs in planning into the seven categories: cultural, political, legal, organizational, knowledge-related, and financial and technology-related factors. We find the framework useful for structuring the discussion, although the factors are sometimes partly overlapping and are not mutually exclusive.

Financial factors relate to the funding for SDG implementation activities. This includes ensuring that enough resources and capacity are available for planning, data collection and data analysis. Technological factors are related to having available the necessary technological solutions and tools for, for example, data collection and data analysis. Knowledge-related factors are related to knowledge about how to operationalize the SDGs, measure sustainability, collect and analyze data, etc. Political factors are related to having the support of democratic institutions at the national, regional or local governmental levels, or from organized interest groups. Cultural factors ensure that policies implemented do not conflict with norms and values, and therefore lack public and/or stakeholder acceptance. Planning cultures might also act as a powerful barrier to new ways of thinking and therefore need to be adjusted [25]. Organizational factors concern issues related to the collaboration within and between institutions. Legal factors are factors that help integrate the policies within existing laws and regulations, ensuring that they do not counteract each other.

We find that a considerable amount of literature already exists on factors facilitating the implementation of the SDGs. We have summed up some of the findings in the literature in Table 1. The publications both include peer-reviewed articles and reports. As the table shows, there are examples of facilitating factors within all the seven categories defined above. However, despite the large amount of literature on factors facilitating the implementation of the SDGs, scientific literature from the Nordic and Norwegian contexts is scarce.

In Table 1 we have also indicated which methodological approaches the literature draws on. Some of the literature provides new empirical knowledge collected through interviews, surveys, and document studies. However, many of the articles and reports are theoretical, some in the sense that they discuss empirical knowledge collected by others. Although the list of literature is not extensive, it indicates an overweight of theoretical or discussion papers and reports. Empirical literature from the Nordic countries include Gassen et al. [26] and SWECO [27].

Our article builds on the existing literature and provides extended and updated empirical knowledge on facilitating factors for implementation of the SDGs in the Norwegian context.

**Table 1.** Identified key factors for successful implementation of the sustainable development goals (SDGs) in local and regional planning.

| Category | Key Factors Facilitating the Implementation of the SDGs | Examples of Literature and Methodological Approaches |
|---|---|---|
| Financial | Provide sufficient resources for planning, data collection and data analysis<br>Finance strategic activities such as workshops, campaigns and education | Gassen et al. [26]—Interviews<br>Satterthwaite [10]—Theoretical<br>Smoke [28]—Theoretical<br>SWECO [27]—Interviews and survey<br>UCLG [29]—Theoretical<br>UN Department of Economic and Social Affairs [30]—Document studies<br>Wymann et al. [31]—Interviews |
| Technological | Make available relevant, disaggregated, high-quality data that permits comparisons with other local and/or regional authorities<br>Avoid using too many indicators<br>Make available tools for data collection and analysis of data<br>Encourage technology and innovation that positively contribute to the implementation of the SDGs | Lucci [32]—Theoretical<br>Mischen et al. [33]—Literature review<br>Nordtveit [34]—Document studies<br>Patel et al. [35]—Document studies<br>SWECO [27]—Interviews and survey<br>UN Department of Economic and Social Affairs [30]—Document studies |
| Knowledge and plan processes | Increase competence on data collection and analysis<br>Increase knowledge about how to work with the SDGs<br>Develop plan processes that deal with conflicting considerations and ensure broad participation<br>Use bottom-up approaches that ensure anchoring in local realities<br>Ensure open and inclusive processes<br>Use methods that engage stakeholders<br>Give room for experimentation, trials and failures<br>Ensure mechanisms that hold societal actors responsible for decisions, investments and actions<br>Perform a cost analysis of the implementation of the SDGs<br>Share good examples and solutions to inspire others | Bowen et al. [36]—Theoretical<br>Gassen et al. [26]—Interviews<br>Hofstad & Vedeld [37]—Survey, document studies and interviews<br>Leal-Arcas [38]—Theoretical<br>Moallemi et al. [39]—Theoretical<br>Satterthwaite [10]—Theoretical<br>Slack [40]—Theoretical<br>SWECO [27]—Interviews and survey<br>UCLG [29]—Theoretical<br>UN Department of Economic and Social Affairs [30]—Document studies |
| Political | Inclusive and representative decision-making at all levels<br>Trust-building between inhabitants and authorities through dialogue<br>Clear communication of national priorities and activities in Agenda 2030<br>Political support for the work with the SDGs | Awan [41]—Literature review<br>Gassen et al. [26]—Interviews<br>Oosterhof [42]—Theoretical<br>UCLG [29]—Theoretical |
| Cultural | Awareness about the SDGs among stakeholders<br>Promote culture as driver for development<br>Relate the SDGs to local activities | Fleming et al. [43]—Interviews<br>Gassen et al. [26]—Interviews<br>Tjandradewi & Srinivas [44]—Theoretical<br>UCLG [29]—Theoretical<br>UN Department of Economic and Social Affairs [30]—Document studies |
| Organizational/institutional | Involve all local authority departments<br>Integrate the SDGs into key steering documents, plans and processes<br>Involve the local population and encourage young people to participate<br>Support sustainable businesses and organizations<br>Form strong partnerships between different local authorities, inhabitants, businesses and voluntary organizations<br>Engage existing partners in long-term commitments for the SDGs<br>Promote collaboration between sectors at all levels<br>Better integration and coordination of management systems between various levels of authority<br>Integrate the SDGs in the institutions' mandates<br>Efficient, transparent and responsible institutions | Bhattacharya et al. [45]—Literature review, document studies, interviews<br>Fenton & Gustafsson [46]—Literature review<br>United Nations [47]—Empirical, documents studies<br>Garcia-Alaniz et al. [48]—Theoretical<br>Gassen et al. [26]—Interviews<br>Hofstad & Vedeld [37]—Survey, document studies and interviews<br>Klopp & Petretta [49]—Theoretical<br>Lucci [32]—Theoretical<br>Oosterhof [42]—Theoretical<br>Shulla et al. [50]—Survey<br>Slack [40]—Theoretical<br>UCLG [29]—Theoretical<br>Valencia et al. [51]—Pilot study<br>Veldhuizen et al. [52]—Theoretical |
| Legal—laws and regulations | Formalizing commitments<br>Adopting buyer requirements<br>Establishment of new procurements deals<br>Legislation securing the implementation of the SDGs (e.g., legal equality, equal right to education) and not counteract it | Awan et al. [41]—Literature review<br>Biermann et al. [53]—Theoretical<br>Gassen et al. [26]—Interviews<br>Gladun [54]—Context analysis, interviews<br>Mokoena & Jegede [55]—Theoretical |

### 3. Materials and Methods

This article is based on data collected for a project financed by the Ministry of Local Government and Modernization [9]. We have used a mixture of qualitative and quantitative data and methods as further described below.

#### 3.1. Survey

An electronic survey was sent by e-mail to professionals working with planning and/or environmental issues in all Norwegian municipalities (356). In the small municipalities, only one person received the invitation to participate, while in the larger municipalities, two or more persons were invited. In total, 715 persons received the e-mail, whereas 132 persons completed the survey, which gave a response rate of 18.5 percent. Although this is not so high at the individual level, in total, 30 percent of the municipalities were represented among the respondents. The represented municipalities showed good coverage in geographical location and size.

The survey included 37 questions with predefined alternative answers, and various themes related to the implementation of the SDGs in municipal planning. In two of the questions, the respondents were directly asked about potential barriers related to the implementation of the SDGs.

The first question was as follows:

1.  To what degree do the following represent a barrier for using the SDGs as a planning tool in your municipality?
    a.  Lack of knowledge about the SDGs in the municipality
    b.  Different understanding of sustainability in various parts of the municipal administration
    c.  Lack of relevance of the SDGs for local planning
    d.  Lack of time/resources
    e.  Lack of methods and tools for using the SDGs
    f.  Lack of guidance from the county council
    g.  Lack om guidance in Norwegian
    h.  Existing guidance is too comprehensive and complicated
    i.  Lack of good indicators for monitoring status and progress.
    j.  Lack of coordination and dialogue across sectors in the municipality
    k.  Lack of political anchoring
    l.  Lack of engagement in the municipal administration
    m.  Lack of engagement among inhabitants in the local community.

The respondents were asked to rank the barriers on a scale from 1 to 5 where 1 represented "to a very small degree" and 5 represented "to a very large degree". The respondents were also asked to comment and give their thoughts about how each of the barriers could be overcome by writing in open text boxes.

The second question was:

2.  Describe other potential barriers and their importance.

This question was followed by an open text box for the respondents to describe barriers and their importance in their own words.

In the open text boxes, many respondents provided rich descriptions of challenges and success factors related to the implementation of the SDGs, which have been valuable in the data analysis.

#### 3.2. Interviews

Interviews were performed with key informants from six municipalities (Ålesund, Narvik, Gloppen, Lunner, Asker, and Arendal) and five county councils (Viken, Nordland, Møre og Romsdal, Vestland, and Agder). Common to all is that they have started to implement the SDGs in their planning, but they differ when it comes to size, geographic location, population, degree of urbanity and rurality, and whether they had recently merged

with other municipalities/counties. The five county councils cover the six municipalities, which allows a multilevel analysis. The municipality of Ålesund is located in the county of Møre og Romsdal, Narvik in Nordland, Gloppen in Vestland, Lunner and Asker in Viken, and Arendal in Agder. Figure 1 shows a map of the location of the case counties.

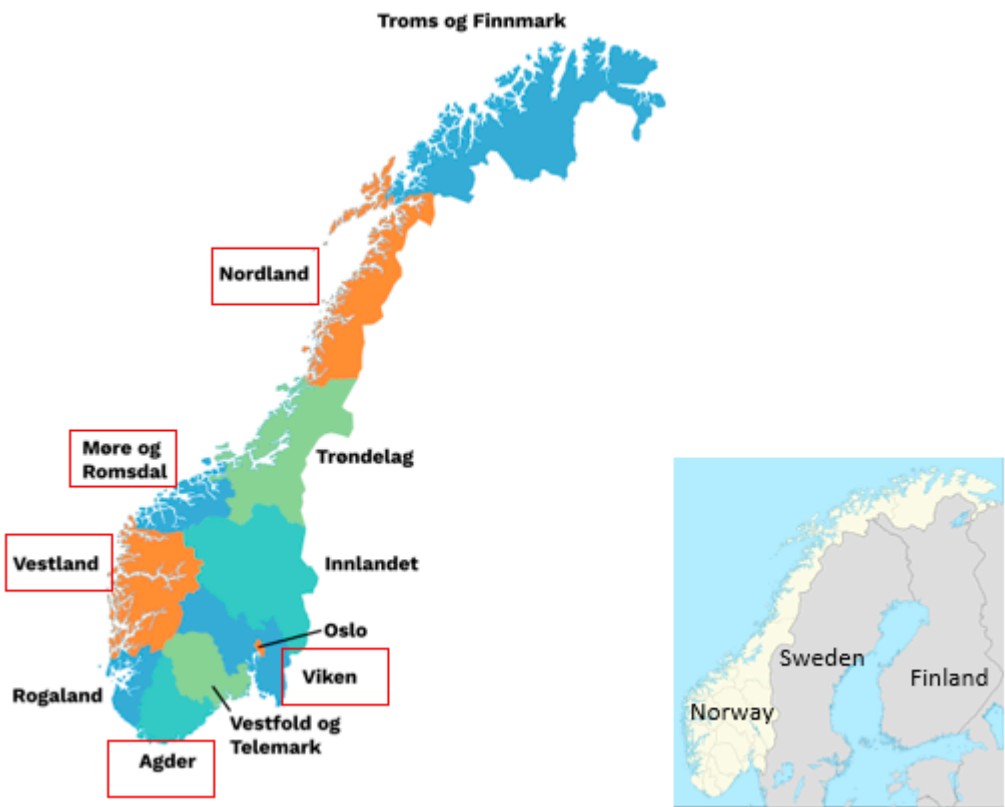

**Figure 1.** Map of Norway showing the location of the case counties (highlighted in the red boxes).

In total, 16 key informants were interviewed. In four of the interviews, two informants participated. The informants all had central roles in implementing the SDGs in planning in their respective municipalities and counties. Several of the informants had management roles in planning, such as head of the planning department, and they had been involved in both overall societal planning and spatial planning.

The interviews were conducted as semi-structured interviews, guided by an interview guide, but at the same time giving the informants room to add relevant information and comments. Eight of the interviews were performed on Skype, one by telephone, and two face-to-face with the informants. The interviews lasted from 50 to 90 min. All interviews, except for the telephone interview, were recorded, and notes were taken both during the interviews and afterwards on the basis of the recordings. In the presentation of findings, the informants are anonymized.

*3.3. Data Analysis*

The notes and transcripts from the interviews were first analyzed by thematic analysis [56]. The data were coded for themes, and a theme was identified when the coder noticed something in the data reflecting the research questions and themes of interest in a patterned way. The informants were allowed to read and comment on the written reports from the analysis of each interview. This was done to ensure that our interpretation of the interviews was in line with what the informants had meant and to allow for corrections of misunderstandings. Several of the informants added information which further enlightened the research questions. Next, both the data from the interviews and surveys were analyzed by the theoretical framework on facilitating factors for implementation of policies

described in Section 2. This includes categorizing the facilitating factors into cultural, political, legal, organizational, knowledge-related, financial and technology-related factors.

The analysis was carried out in four steps. First, we analyzed data as part of the project financed by the Ministry of Local Government and Modernization [9]. Second, from the data, we subtracted issues with particular relevance for facilitating factors. Third, the semi-structured interviews, survey data, and theoretical literature allowed us to analyze our findings from different angles and hence triangulate data [57]. Finally, we analyzed the data by organizing them into the predefined categories that our theoretical framework provides.

## 4. Results and Discussion

A common experience among the respondents and key informants in the survey is that working with the SDGs has created enthusiasm and has been useful, important, and exciting. Several informants also mentioned that they had learned a lot in the process. However, working with the SDGs could be challenging in various ways. In this section, we present and discuss the respondents' and key informants' thoughts and experiences about factors hindering the implementation of the SDGs in local and regional planning, and what they think may be key factors in overcoming these and facilitating the implementation process.

The discussion is organized in accordance with the three research questions. In Section 4.1, we present and discuss identified factors facilitating the implementation of the SDGs in Norwegian local and regional planning, while Section 4.2 discusses how the factors are conditioned by the different Norwegian local and regional contexts and how they differ from those in the international literature.

### 4.1. Identified Factors Facilitating the Implementation of the SDGs

In Table 2 the identified facilitating factors have been categorized by type. The categories of factors are discussed in their respective sections below the table.

#### 4.1.1. Financial—Capacity and Resources for Development Work

Lack of capacity and resources was mentioned as one of the most important barriers for implementing the SDGs in local and regional planning by the respondents to the survey. This comment from a respondent illustrates this well: "Daily operations 'eat up' time and capacity for development work". Particularly the smaller municipalities pointed at the need to rely on the work of others, since they experience it as challenging to manage to do development work themselves. This often led to copying from larger municipalities with the danger of not being able to adequately consider their specific context. Moreover, they pointed at the fact that it is a huge task for small municipalities to make good plans with the SDGs as a framework when they often only have one or two employees working with planning.

Further, our findings show that there is also a need to have the capacity to become acquainted with the literature on the SDGs and particularly the implementation process. In the interviews and survey, the informants mostly concentrated on reading the guidance material from the national authorities. While about 60 percent of the respondents answered that they were familiar with two specific Norwegian guidelines (see [58,59]), about 70–80 percent did not know about relevant English guidelines such as those by Gassen et al. [26], SWECO [27], and The Global Taskforce of Local and Regional Governments [22,60–62]. A large share (39 percent) of the respondents in the survey, considered a lack of methods and tools for implementing the SDGs as a large or very large barrier. However, this may not only be because the methods and tools are lacking, but rather also a question of making existing tools accessible and having the time to explore them.

The results clearly indicate that successful implementation of the SDGs in local and regional planning requires that both the municipalities and counties have the capacity and resources needed to work with SDG implementation. In order to prioritize time for working with SDG implementation, respondents mentioned the significance of administrative leaders expressing that this was an important task with high priority. Inter-municipal

collaboration was also mentioned as a potential strategy to overcome the capacity barrier. However, as one of the respondents commented, the work with the SDGs needs to be integrated into existing activities, not become something separated from the service provision activities for which the municipalities and county councils are responsible.

**Table 2.** Facilitating factors identified in the survey and the interviews with key informants.

| Category | Key Factors Facilitating the Implementation of the SDGs in Norwegian Local and Regional Planning |
|---|---|
| Financial—esources and capacity | Allocate time and capacity for development work<br>Allocate time for getting familiar with literature and guidance material |
| Technological | Access to adequate methods and tools for implementing the SDGs<br>Access to guides written in Norwegian<br>Access to indicators for measuring status and progress on sustainability—particularly how to consider qualitative issues not measurable to ensure that they are not left out<br>Access to guides on how to operationalize the goals locally<br>Access to guidance materials and information relevant for small urban municipalities |
| Knowledge | Necessary to have access to comprehensible and relevant knowledge about the SDGs and which role they should play—need for a systematic representation of the overwhelming literature<br>Access to knowledge about relevant networks that exist<br>Access to good examples on implementation of the SDGs in local and regional planningHave systems for sharing knowledge within and between organizations |
| Political | Need for clear messages from the national government of what they want and expect<br>Need for engagement in the municipality and county council organizations and the local communities<br>Necessary with good anchoring and support from regional and local politicians<br>The SDGs must be incorporated in the financial plans<br>Avoid "green washing" of existing activities instead of change<br>Avoid budgets being tied up with statutory tasks and earlier priorities |
| Cultural | Need for a common understanding of what sustainable development means and what the work with the SDGs means and how to interpret the SDGs locally and regionally |
| Organizational/institutional | Need for internal coordination and dialog across sectors in the municipality/county council<br>Need for common methodology for working with the SDGs across levels<br>Need for a cross-sectorial understanding of the goals at the national level<br>Need for consistency in the state authorities' principles, directions and guidelines at underlying levels, particularly within spatial and transport planning<br>Avoid work with the SDGs being a top-down process not ensuring involvement by those who are going to implement the goals.<br>Need for a good connection between community plan and financial plan<br>Need for county councils to give guidance and help to municipalities<br>Need for coordination of activities across local, regional and national levels<br>Need for arenas for collaboration with actors such as businesses and academia |
| Legal—laws and regulations | Use the opportunities lying within the Planning and Building Act |

It has to be mentioned that the United Nations Institute for Training and Research (UNITAR) has developed activities, courses, webinars and conferences, aiming to help national, regional and local authorities to build the capacity to implement SDGs and monitor progress. In addition, the SDG Accelerator and Bottleneck Assessment tool is aimed at supporting countries in identifying policies and measures that can help to solve bottlenecks and accelerate the implementation of the SDGs [63]. Nevertheless, our findings show that these guidelines are not widely known or used among municipal planners in Norway at the moment.

### 4.1.2. Technological—The Need for Accessible Methodologies, Tools and Indicators

Many of the informants experienced a lack of (knowledge of) suitable methods, tools and indicators for implementing the SDGs and difficulty in relating the tools and indicators to their local context. One-fourth of the respondents in the survey replied that guides that are too complex and extensive were a barrier to a large or very large degree, and approximately the same amount answered that the lack of guides in Norwegian was a barrier. They also expressed that the international literature was not always relevant for the Norwegian context. As one respondent in the survey commented: "Guides in English

are experienced as far away from our daily lives. Norwegian guides focus on Norwegian contexts and that feels more relevant".

Another point that was mentioned was that for smaller municipalities, the existing literature and guides on SDG implementation were seen as not being that relevant since the examples often came from bigger municipalities with specifically urban challenges. A respondent in the survey made the point clear:

> The large themes concerning environmental issues in planning are more targeting the larger cities and their problems. In our case it is the scattered settlements, small business areas and issues related to this, for which finding solutions is urgent. Safeguarding green spaces, handling surface water and densification, are minor problems for us. This means that the existing guides are not so relevant for us.

A similar comment was made from another respondent in the survey, supporting the impression that the smaller municipalities do not experience that existing guidelines fit their situation to the same degree as the larger ones:

> I receive e-mails from the Ministry and others and have noticed that reports and seminars (often international) exist, however often of the type 'sustainable cities'. We have not considered these as relevant for a small rural municipality as ours, which barely has small villages.

Some respondents to the survey argued that it ought to be a task for the Ministry to find "best practices" that the municipalities could use: "The municipalities do not need to be unique in everything, and the politicians in different parts of the country do not disagree on everything", as one respondent in the survey commented.

One municipality had good experiences with using a "significance analysis" for making the goals relevant in their local context. Through the "significance analysis", goals and targets were systematized on the basis of which goals the municipality could influence directly and the areas that the municipality ought to take direct responsibility for in the work of developing the local community and services for the inhabitants.

On the one hand, there seems to be a wish for clear methods, indicators, and systems for the measuring and reporting of effects in such a way that it is possible to make comparisons with others. On the other, there is also a recognition that not everything can be measured. Several informants pointed at the need to know about status, in order to measure progress and in some way "put a number" on the efforts made; however, it is not possible to capture in the existing indicators. Several areas were mentioned which were difficult to describe due to the lack of data. In addition, there are themes that the SDGs and their targets themselves do not capture very well. As examples of this, the preservation of cultural heritage and the conservation of nature were mentioned. The management of marine resources and agriculture were also mentioned as not being handled properly in the SDGs. This shows that although the SDGs seem to embrace broadly, there are still themes that are not sufficiently covered.

If too much focus is on those issues that can be measured, there is also a danger of leaving out important areas. The informants therefore considered it essential to make subjective judgments in addition to quantitative measurements, and perhaps it was not necessary to produce more indicators but rather to develop methods and tools for considering issues that cannot be measured. As stated by one informant in one county council:

> Not all we want . . . can be measured. You can measure depopulation and the occurrence of muscle and skeleton diseases, and you can ask people if they are depressed. However, youths leaving high school is a complex issue... There is a reason behind the leaving, which is complex. So, in my opinion, it is important that we measure what we are able to measure, but then it is extremely important to obtain a subjective consideration of what you really want, where you can add another dimension.

One informant from a county council participating in the U4SSC network [64] underlined the need for them to consider key performance indicators more relevant from a regional perspective, and that not all the indicators developed within the network was necessarily suitable for them. To help with this, it was necessary to develop a digital tool. On the one hand, it is necessary to operationalize the goals locally, and some of the informants expressed that this was a demanding task. On the other, several informants argued that it is important to make use of what methods and indicators already exist, and that one has to make sure that not too many resources are spent on establishing unique reports on the SDGs. One municipality used external expertise to get started with implementing the SDGs but then took over the control of the process and experienced that this increased the municipality's ownership of the matters in question.

In several of the interviews, the relationship between a holistic approach and a more selective approach to the SDGs was problematized. While there is a need for a holistic approach in order to capture the whole picture and the dynamics between goals, some emphasized the need to narrow down the work and prioritize between the goals; otherwise, it would be too overwhelming and difficult to get started. These two approaches are very different.

### 4.1.3. Knowledge—Internal and External Knowledge about the SDGs

As already mentioned, a lack of knowledge about the SDGs and existing methodologies, guides and indicator tools represents an obstacle to the implementation of the SDGs. Attending courses, seminars and workshops were mentioned as strategies that could increase knowledge of the SDGs and existing materials. "We at least need to talk about it," as one respondent in the survey commented. However, several recognize that increasing knowledge and competence takes time and requires continuous focus in many arenas.

While all municipalities, independent of size, reported using the Norwegian Association of Local and Regional Authorities (KS) as a source, the larger municipalities responded to using the United Nations Association of Norway and Norwegian Smart Cities to a greater degree compared to the smaller ones. The informants mentioned participation in networks as being useful for obtaining knowledge about SDGs, tools and indicators, and sharing experiences from implementing the SDGs in planning. However, it was mentioned that it is challenging to gain an overview of all the networks that exist.

In order to make the existing literature on the SDGs and their implementation more accessible, there is both a need to systematize the literature and to translate the findings to the local (here Norwegian) language and contexts, as mentioned in the previous sections. As one informant commented:

> There is no doubt that a lot of information exists, however you drown in all the information and do not know how to use it . . . There is a need for a clear guide that thematically approaches how to use the information, both in a simple way and more advanced. It also has to give guidance on where the information can be used most effectively—in the municipality plan and its regulations, or in zoning plans and associated regulations, or both.

Another respondent to the survey commented:

> As planners in small municipalities, we have many roles and tasks . . . It is difficult to have enough time to familiarize oneself with new knowledge.

This illustrates that there is need to establish routines and systems for sharing information and competence within the municipalities and county councils. It is also seen as crucial that regular employees get involved in the implementation process to ensure that it becomes part of the daily work throughout the organizations and not only something going on in specific parts of the administration.

### 4.1.4. Political—Commitment among Politicians, Stakeholders and Inhabitants

All informants agreed that anchoring and support from the politicians is crucial for successful implementation of the SDGs locally and regionally. As one informant from a municipality commented:

> I wish sometimes that they were more concrete and visionary, because it is much more difficult for the administration to be visionary. The latter implies going beyond your mandates, promoting a case no one has asked for. It is much easier the other way around.

Although the informants reported that many politicians put the SDGs high on the agenda, work still needs to be done to achieve a broad commitment across parties and to reduce the polarization of the debates regarding the SDGs. One respondent from a county council argued that it was important not to depoliticize the SDGs and make such rigid systems that have all priorities set out in advance. There had to be room for prioritizing, the informant argued, but it was important to establish a good decision base so politicians were able to see the consequences of their decisions. However, several emphasized that the SDGs must be something more than just a checklist of which goals, various measures and decisions contribute to achieving. One barrier mentioned was that statutory tasks and earlier priorities often reduced the room for action in budgets, as pointed out by one respondent in the survey:

> One important barrier lies in the budget being to a large degree tied up with statutory tasks and earlier priorities. You need time to turn this around.

It was mentioned as important to provide arenas for participation and knowledge-sharing with politicians (e.g., meetings, seminars, and workshops) to increase politicians' knowledge about the SDGs. This was also suggested as a good strategy to increase awareness and commitment to the SDG work internally in the municipalities and county councils. In addition, it was mentioned as important to be able to move beyond just talking about the SDGs. It is important to be able to show some results from the work, either internally, or good examples from "first movers". Use of public meetings and increased communication of knowledge about the SDGs, and results from working with them, were mentioned as factors which could increase awareness and engagement among inhabitants. However, it was recognized that differentiated measures needed to be implemented in order to reach various groups of the population.

Twenty-eight percent of the respondents also considered lack of engagement and involvement throughout the organization in the work with the SDGs as representing a barrier to a large or very large degree. They call for better internal communication, collaboration and awareness-building about the SDGs and sustainable development throughout the municipal administration. It was considered a challenge to be able to involve all politicians, and it was recognized that it was important to remember that not all politicians have the same knowledge about the SDGs. Thus, extensive information activities are necessary to enlighten both politicians and employees in the municipality and county council administrations. One informant emphasized the importance of associating what is done in the municipal organization with the rest of the local community. It should be noted that several of the municipalities in the study had performed various, partly innovative, co-creation activities in the work with the municipal plan in order to ensure support and commitment from all stakeholders.

Several informants were concerned about making politicians aware of how various political decisions impacted the SDGs. One of the county council interviewees highlighted striving for economic growth as "the elephant in the room". It was questioned to what degree the SDGs contributed anything new to the discussion about societal development, in line with the objectives of the Planning and Building Act (PBA) that planning should contribute to sustainable development. Several were worried that the SDGs would only lead to "green washing" of existing politics. As one informant from a municipality commented: "I am a little bit afraid that the SDGs are associated with everything you do and

do not lead to any changes". The informant highlighted it as important to make sure that the SDGs impact priorities in practice, and not only become "a new wrapping".

A key issue that informants emphasized in the interviews was that although the work with the SDGs is long term, it is also important to show that things are happening now, and that the involvement of inhabitants also commits the municipalities to take the issues raised by inhabitants seriously and act upon them, although they may not relate to the specific plan in question.

### 4.1.5. Cultural—A Common Understanding of Sustainability

The survey revealed that 33 percent of the respondents consider different perceptions and interpretations of sustainability in the municipal administrations as representing a barrier for SDG implementation to a large or very large degree, and that it was challenging to secure social, environmental, and economic sustainability simultaneously.

Most informants agreed that participation and involvement of a broad spectrum of stakeholders is a key factor for successful implementation of the SDGs. However, during the interviews, challenges related to dialog and collaboration with stakeholders were thematized, as people interpret the SDGs differently and want different things from them. As one informant from a county council commented:

> Everyone is working with the SDGs; however, they have implicit objectives that may be diverging ... If you don't uncover these implicit objectives, then you are not talking about the same things when you meet up.

These challenges assert themselves across administrative levels and between different actors, illustrating the need for a common understanding of what the work with the SDGs means.

In order to develop a common understanding of sustainability among stakeholders internally in the municipalities and county councils, the respondents suggested several potential measures such as increased collaboration and communication across sectors, arranging of shared meetings and projects, and shared competence-building programs across sectors. Anchoring and commitment in the top management of the municipalities and county councils were seen as crucial for this to happen. Manager development programs incorporating the SDGs were also mentioned as a measure which could help develop a shared understanding of what sustainability means and how it impacts the work of the municipalities and county councils.

### 4.1.6. Organizational—The Importance of Cross-Level and Cross-Sectorial Work and a Coherent Goal Structure

The different national authorities also have a key role in facilitating the implementation of the SDGs locally and regionally by coordinating their activities and following their own principles, directions and guidelines at all levels and sectors, and in all their meeting points with the local and regional level.

Since the SDGs are cross-sectorial by nature, one municipality pointed to the importance of developing cross-sectorial priority areas where different service areas are working with the same goals. The same municipality also stated the importance of building a "red thread" from the overarching goals in the municipal plan, to the more concrete strategies and measures in the subordinated plans. In particular, the connection to the financial plan was pointed out as important, because then you had to make specific priorities. A direct connection between the community plan and the financial plan will enable reporting against the financial plan which can be directly associated with the goal structure at the base of the community plan. The SDGs then become a part of the financial plan, and the community part of the municipal plan will become more useful, a respondent argued.

Several mentioned the involvement of and co-creation with stakeholders as important when developing plans, by contributing to anchoring the plan among stakeholders and developing the plans based on what stakeholders find important and relevant, as well as making it easier to achieve commitment to the plans across sectors. "Sustainability

breakfasts" had successfully been used by one county council to make the SDGs better known internally in the organization and create engagement in sectors other than the planning sector.

Twenty-eight percent of the respondents from the municipalities thought that lack of guidance from the county councils represented a barrier for implementation to a large or very large degree. Some commented that they were waiting to become part of the county council's development work and that guidance from the county councils was particularly important for smaller municipalities with few employees dedicated to working with SDG implementation. However, the interviews with the representatives from the county councils suggested that several found it challenging to provide guidance to the municipalities because they were themselves trying to figure out what to do. It was recognized that small and large municipalities may have different needs regarding guidance from the county councils. Although the situation may be more transparent in smaller municipalities, the competence and capacity to work with SDG implementation may be scarcer here, compared to larger municipalities.

The interviews revealed that there is a need for good examples illustrating the variety of approaches and methodologies of working with the SDGs that have been tested and of which experience has been gained.

It was pointed out as important to get started with implementing the SDGs and that much could be learned along the way by trial and error. Several of the interviewees also emphasized the importance of work done by enthusiasts and inspiration gained from networks and other actors in order to get started. A good strategy may be to give room for enthusiasts, participate in networks, and make allowances for trial and error. One small municipality specifically mentioned the regular meetings in a planning forum as important. It helped them to shift their focus from daily operations to important community issues.

### 4.1.7. Laws and Regulations—The Planning and Building Act as Tool for SDG Implementation

A perception among most of the informants was that the Norwegian Planning and Building Act (PBA) (https://lovdata.no/dokument/NL/lov/2008-06-27-71 accessed on 15 January 2021) is well suited to serving as the framework for a cross-sectorial and holistic approach to the SDGs. The informants also thought that the Planning and Building Act would contribute to a coordinated approach to the SDGs across the local, regional and national levels. In planning, working with holistic societal development and coordinating various stakeholders and interests is nothing new. However, the informants from the municipalities stated that the SDGs had not made this work any easier and that a more exciting dialog had emerged between the municipalities and other stakeholders in the community. The SDGs, in a way, serve as a common language across subjects and sectors. The county councils reported that the SDGs had contributed to making the county councils' broad spectrum of responsibilities visible in society, and subjects that had not been prioritized earlier were now on the agenda. Some of the informants also thought that the SDGs could contribute to highlighting and finding solutions to conflicts between goals, and to shedding light on political dilemmas.

The implementation of the SDGs in planning concerns many actors in and around the municipalities and county councils, and therefore requires systematic efforts from many actors over a long period of time. One respondent argued for it to be a prerequisite that the PBA be used as an active management tool in the process.

### 4.2. The Impact of the Norwegian Local and Regional Context

In this section we discuss the identified factors presented in the previous section in relation to the two research questions: (1) how are the facilitating factors conditioned by the different local and regional institutional contexts, and (2) how do these factors from the Norwegian context correspond to or differ from those in the international literature?

Although countries and regions may be geographically, culturally, and economically different, they seem to struggle with many of the same issues when addressing the SDGs

in their respective contexts. The size of the municipality administration seems to be more important than geography in explaining the differences observed. This is not surprising giving that small and rural municipalities often have limited planning capacity and smaller professional environments [17]. Our findings indicate that this affects how the municipalities can relate to the national guidelines and implement the SDGs in their planning. However, realizing Agenda 2030 depends on the ability of both large and small municipal administrations in urban as well as rural areas to initiate sustainable transition processes. Thus, the literature needs to further develop the understanding of the factors enabling implementation of the SDGs in different institutional contexts, and how cross-sectorial synergies can be achieved in these different institutional settings. Small municipalities have challenges different to the vast literature on "sustainable cities," and thus, need tools to handle their specific challenges.

When we compare the results from the literature (Table 1) and our findings from the Norwegian cases (Table 2), we find many similarities. In general, our findings from the Norwegian context seem to align with experiences elsewhere. A common challenge across different institutional contexts relates to the provision of sufficient resources for planning activities, data collection and analyses. This also seems to be more prominent in smaller municipalities. In our study, 58 percent of respondents in the municipalities with less than 10,000 inhabitants reported that the lack of capacity and resources represented a barrier to a large or very large degree, while only 39 percent of respondents in the larger municipalities answered the same. The same trend was also seen regarding knowledge about the SDGs in the municipalities. Here, 42 percent of the respondents in the smaller municipalities said that this represented a barrier to a large or very large degree, while only 26 percent said so in the larger municipalities. An interesting finding is that, for reasons unknown, the smallest municipalities with less than 1500 inhabitants departed from this trend both with regard to capacity, resources, and lack of knowledge being barriers. A reason might be that often fewer stakeholders are involved in smaller municipalities. The municipal administrations are also smaller and the variation in activity in the municipality is often limited.

Both in Norway and other countries, there seems to be a strong need for relevant tools and indicators for data collection and analyses, monitoring of sustainability, and guidelines for implementing the SDGs in the planning processes. However, the lack of such methods expressed by the informants in the Norwegian context seems to be just as related to making existing tools and guidelines available for Norwegian-speaking planners in a more comprehensive manner. The extensive literature, partly consisting of large-scale reports, need to be systematized, translated, and made relevant probably not only for Norwegian contexts, but for other regional and local contexts as well. The extensive work by, for example, the UN is in practice difficult to use for local and regional governments, and even more so for smaller municipalities than larger ones. The interviews indicate that the smaller municipalities may need more guidance from the regional authorities and could particularly benefit from making the existing methodologies, guides and indicator tools more widely available. In addition, they may also benefit from the development of "best practice" methodologies, so that each and every little municipality does not have to do all the development work by themselves.

Both in the Norwegian context and in the international literature, there is a need to ensure the anchoring of the SDGs at all levels and with stakeholders within the community. The informants in our study emphasized the need to develop a common understanding among stakeholders of what the work with the SDGs actually means. They also highlighted the polarizing debate between those skeptical to climate change and their opposites as challenging for the implementation of the SDGs. The need to involve the local population was also highlighted in the literature, in addition to the participation of young people being seen as important.

Both the literature and this study point to the importance of institutional factors, such as better coordination of plans, activities and processes across different sectors, actors and

government levels. Clear communication of national priorities was seen as important. The Norwegian government has yet to take a stand on any national priorities, leaving this in the hands of regional and local governments. On the basis of the literature review and our findings, the formulation of a national agenda seems crucial to successfully implementing the SDGs. In the Norwegian context, the role of the national authorities was highlighted as important, especially the need for the state to follow up its own principles, directions and guidelines at underlying levels. On the one hand, the national authorities call for the SDGs to frame all planning activities, while on the other hand, they use other measurement tools, which are not based on the SDGs, to prioritize the financing of activities such as developing transport infrastructure. In order for the SDGs to have a real impact, the national authorities need to integrate the SDGs in the work performed by the various underlying levels as well.

One interesting finding in the Norwegian context, which we did not find in the literature, was the usefulness of existing legal instruments, most specifically the Norwegian Planning and Building Act (PBA), in implementing the SDGs. While the literature pointed to a need for legislation to secure implementation, our findings indicate that the formal legal framework in Norway is, in general, considered appropriate for implementation. There is broad agreement among the informants in our study that the PBA may serve as a good framework for implementing the SDGs. This is a tool which has already integrated the focus on sustainable development where the SDGs may be seen as an operationalization of this. Using an already institutionalized framework for the implementation of the SDGs may prove easier than developing entirely new ones.

There were also fewer mentions of issues such as a need for efficient, transparent and responsible institutions in the Norwegian context compared to the international contexts. The same goes for a need to ensure mechanisms that hold societal actors responsible for decisions, investments, and actions. This seems to indicate a high degree of trust in institutions amongst our informants and respondents. Lundberg et al. [9] found that SDG 16, which relates to strong institutions, is prioritized to a lesser degree than many of the other goals amongst Norwegian municipalities. This might also be interpreted as a sign of relatively well-functioning institutions. Considering the strong institutions and the appropriate legal framework, as noted above, we might question how far the Norwegian effort to achieve the SDGs has come. At the same time, sustainable development has been a topic in Norwegian planning for several years, and even though measures might not be linked directly with the SDGs, they might, in practice, cover these issues. This is perhaps in opposition to other places where the SDGs are seen to bring something substantially new to local planning and policymaking.

## 5. Conclusions

In this study, we have examined the barriers that challenge the implementation of SDGs in planning at the local and regional level in Norway, and how municipal and regional planners perceive them in their practical work. Relating these to the framework of facilitating factors, we contribute empirical insights into how these challenges can be handled. The SDGs are implemented in very different institutional contexts and, as we have shown, this affects how barriers are perceived and what facilitating factors are needed to enable a successful implementation. Knowledge about these factors, and measures that can be taken to improve them to enable capacity-building at the local and regional level is vital. In practice, financial, technological, knowledge, political, cultural, institutional, and legal aspects will affect planning and planners' ability to implement the SDGs.

The expectations on local and regional planning in addressing the SDGs in Norway are high, and municipalities and counties play a key role in the Norwegian efforts to realize Agenda 2030. However, as mentioned in the introduction, the Auditor General [18] has criticized the lack of national coordination in the implementation of the SDGs. Our study shows that there is a need for a clearer voice from the national authorities about what this means in practice for local and regional authorities: what the challenges and objectives are. There is a demand for better knowledge and competence in how the SDGs

can be implemented locally and regionally. This can be responded to by systematizing the existing literature on tools and guidelines, by making them accessible to planners, and by encouraging participation in existing networks and establishing new ones to enhance the sharing of experiences and learning.

There are reasons to believe that for the SDGs, existing planning tools may be both a promising route and a barrier. The findings in this study indicate that the Norwegian Planning and Building Act is an example of the former. A suggestion for further studies would be to do comparative studies of similar planning systems in different countries and assess their potential as tools for implementing the SDGs. This could provide valuable knowledge transfer, improving the implementation of the SDGs in planning at local and regional levels.

If the SDGs are to make a difference through concrete actions, they have to be incorporated into action plans and existing planning tools. This will require collaboration and development work across different sectors and authority levels, as well as the development of guidelines on how this can be done. Further, we would like to emphasize the need to develop a broader understanding and approach to SDG 11, concerning sustainable cities and local communities, to make it relevant for smaller municipalities, often in rural areas. This will likely apply to other European rural municipalities as well. In line with Kulonen et al. [13], who draw on research form remote mountain regions in Europe, we argue for the need to develop approaches and framework that can account for spatial and institutional considerations at the sub-national levels. Smaller municipalities face different challenges to larger ones, and our study indicates that they may need other types and degrees of support when implementing the SDGs. Expanding on their role and opportunities to plan and develop more sustainable communities may be vital for the implementation of the SDGs outside the large urban areas too.

**Author Contributions:** Conceptualization, K.G.B., M.B.R., A.K.L. and M.B.; methodology, K.G.B., M.B.R., A.K.L. and M.B., validation, K.G.B., M.B.R., A.K.L. and M.B.; data analysis K.G.B., M.B.R., A.K.L. and M.B.; investigation, K.G.B., M.B.R., A.K.L. and M.B.; writing—original draft preparation and review and editing, K.G.B., M.B.R., A.K.L. and M.B. All authors have read and agreed to the published version of the manuscript.

**Funding:** This research received no external funding.

**Institutional Review Board Statement:** The study was conducted according to the guidelines of the Declaration of Helsinki, and approved by the Institutional Review Board of NSD (Norwegian Data Protection Service, January 2020).

**Informed Consent Statement:** Informed consent was obtained from all subjects involved in the study.

**Data Availability Statement:** The data presented in this study are available on request from the corresponding author. The data are not publicly available due to privacy restrictions.

**Conflicts of Interest:** The authors declare no conflict of interest.

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
