# Peer review of "Factors Facilitating the Implementation of the Sustainable Development Goals in Regional and Local Planning—Experiences from Norway"

_sustainability, doi:10.3390/su13084282_

Round 1
Reviewer 1 Report
The article deals with an interesting topic and fits for Journal’s aim. The issue discussed in this study is original with reference to contemporary scientific discussion on the implementation of the sustainability paradigms into Norwegian regional and locak planning contexts. The study highlights remarkable points of view and critically mention issues in the definition and applicability of Sustainable Development Goals.
In my opinion, the study should not be accepted for publication in its present version since it presents a number of caveats. In a reviser version of the papier, the author should carefully address the following points.
- Introduction should inform about the research design and hypotheses of manuscript.
- The paradigms of sustainability is key-concept of the paper. However, there is no way to detect which is the author’s conceptual approach to this paradigm.
- The literature review should contain references concerning the methodology used in the study. Moreover, I would suggest the author put in evidence analogies and differences of related studies with respect to the methodology which implements in this study.
- Author should present survey form in appendix.
- I would suggest the author make the reader aware of the possible reasons of the results.
- All references should be improved.
- Line 241 should be correct.
Author Response
Thank you for reading our paper and giving constructive and useful comments. We have carefully considered all the comments and below follows a description of how we have addressed them.
Comment 1:
Introduction should inform about the research design and hypotheses of manuscript.
Two sentences informing about the hypotheses of the manuscript is provided in the last paragraph before the research questions (lines 111-115).
A paragraph has been inserted below the research questions (after defining facilitating factors) with a short description of how the data has been collected (128-131).
Comment 2:
The paradigms of sustainability is key-concept of the paper. However, there is no way to detect which is the author’s conceptual approach to this paradigm.
There are many ways to define the concept of sustainability and people use the term in different ways. We have added a paragraph at the beginning of the Introduction defining the concept of sustainable development with a reference to the Brundtland report (lines 26-34).
Both the concept of sustainability and the SDGs could both be discussed and criticized; however, this has not been the aim of this paper. We take for granted that the SDGs have been adopted by among others, Norway, and that the Norwegian government has decided that the SDGs should be the framework for local and regional planning. The aim of the paper is to study the implementation process and factors that hinders and facilitates this implementation.
Comment 3:
The literature review should contain references concerning the methodology used in the study. Moreover, I would suggest the author put in evidence analogies and differences of related studies with respect to the methodology which implements in this study.
We have added information about the methodological approaches in the literature in Table 1. In addition, we have added a paragraph (second above the table) discussing the methodological approaches and highlighting two examples of reports that have provided new empirical knowledge from the Nordic countries through interviews and a survey (lines 191-197).
Comment 4:
Author should present survey form in appendix.
Since we have only used the results from two of the questions in the 37 questions long survey, we do not find it correct to include the whole survey. Instead we have included in section 3.1 a more thorough description of the two questions from the survey we have used. This includes a description of the predefined alternative answers that the respondents were asked to rank (lines 220-244).
Comment 5:
I would suggest the author make the reader aware of the possible reasons of the results.
A more thorough description of the reasons of the results is provided in section 4.2 (lines 618-628).
Comment 6:
All references should be improved.
The whole reference list has been revised and the reference style has been changed to numbered style according to the journal guide.
We have also changed the references to the correct reference styles.
Comment 7:
Line 241 should be correct.
The reference to table 2 has been corrected in the first paragraph of section 4.1 (line 309).
Reviewer 2 Report
This paper present merits for publication in Sustainability journal, but need more work in the areas: theroretical discussion and references. The SGD official literature from Springer Pubs. need quoted and discussed in the paper, and need key references in this field of research from geographical literature, eg Robinson et al...
Besides need a more strong conclusion about the case of study en other european rural environments.....
Author Response
Thank you for reading our paper and giving constructive and useful comments. We have carefully considered the comments and below follows a description of how we have addressed them.
Comment 1:
This paper present merits for publication in Sustainability journal but need more work in the areas: theroretical discussion and references. The SGD official literature from Springer Pubs. need quoted and discussed in the paper and need key references in this field of research from geographical literature, eg Robinson et al...
The SDG literature from the Sustainable Development Goals Series from Springer is highly relevant for the paper. We have chosen to refer to some of this extensive literature that we believe is of particular interest for our paper. In the first paragraph of section 2 (lines 147-157)) we have inserted a reference to Alibašić (2018) and Nhamo et al. (2020). In addition, we have mentioned three references from the book series in the second paragraph of section 1 (lines 43-44).
We agree that the geographical perspective is highly relevant when discussing implementation of the SDGs in regional and local planning, and we have inserted a couple of references to geographical literature in the fourth paragraph of the Introduction (lines 58-66). However, we have in this paper chosen to focus on planning and to study the implementation of SDGs from a planning perspective, where planning covers both small and large municipalities and regions. We have inserted “institutional” in the second research question to underline this. In this perspective, we focus on the institutional dimensions rather than the geographical dimensions. Therefore, we choose not to cover the geographical literature more thoroughly. We are not familiar with the reference “Robinson et al” and need more information about the reference to be able to find it and consider it.
Comment 2:
Besides need a more strong conclusion about the case of study en other European rural environments...
Some sentences have been added in section 5 Conclusion underlining that the arguments apply to other European rural municipalities as well (lines 757-760).
Round 2
Reviewer 1 Report
Congratulations!!!
Good improvement.
Author Response
Thank you. We are glad to hear that.
Reviewer 2 Report
This paper presnt merits for publication but not include several key references from geographical literature in remote rural area. Is an option only work with previous papers in planning and sustainability. But need some explanation and justification in the text. Introduction section.
Author Response
Comment: This paper present merits for publication but not include several key references from geographical literature in remote rural area. Is an option only work with previous papers in planning and sustainability. But need some explanation and justification in the text. Introduction section.
We agree that the paper benefits from a better explanation of our focus on planning in both rural and urban, small and large municipalities and regions, and the institutional context in which the implementation of the SDGs takes place. We have added the number of Norwegian municipal administrations in line 83, and two sentences in the paragraph above the research questions in this revised version of the paper (lines 112-114). We have also changed from “municipalities” to “municipal administrations” in line 115.